# Social influences on physical activity for establishing criteria leading to exercise persistence

Ensela Mema[1], Everett S. Spain[2], Corby K. Martin[3], James O. Hill[4], R. Drew Sayer[4], Howard D. McInvale[5], Lee A. Evans[6], Nicholas H. Gist[7], Alexander D. Borowsky[8], Diana M. Thomas[6]*

1 New Jersey Center for Science, Technology and Mathematics (NJCSTM), Kean University, Union, New Jersey, United States of America, 2 Department of Behavioral Sciences and Leadership, United States Military Academy, West Point, NY, United States of America, 3 Body Composition and Metabolism, Pennington Biomedical Research Center, Baton Rouge, LA, United States of America, 4 Department of Nutrition Sciences, University of Alabama-Birmingham, Birmingham, AL, United States of America, 5 Special Projects Department, The MITRE Corporation, Huntsville, AL, United States of America, 6 Department of Mathematical Sciences, United States Military Academy, West Point, NY, United States of America, 7 Department of Physical Education, United States Military Academy, West Point, NY, United States of America, 8 University of California, Davis, Sacramento, CA, United States of America

☯ These authors contributed equally to this work.
* diana.thomas@westpoint.edu

**Data Availability Statement:** Data will be made available by request after review by the United States Military Academy Chief Data Officer. The review will determine the risks to personnel for

## Abstract

Despite well-documented health benefits from exercise, a study on national trends in achieving the recommended minutes of physical activity guidelines has not improved since the guidelines were published in 2008. Peer interactions have been identified as a critical factor for increasing a population's physical activity. The objective of this study is for establishing criteria for social influences on physical activity for establishing criteria that lead to exercise persistence. A system of differential equations was developed that projects exercise trends over time. The system includes both social and non-social influences that impact changes in physical activity habits and establishes quantitative conditions that delineate population-wide persistence habits from domination of sedentary behavior. The model was generally designed with parameter values that can be estimated to data. Complete absence of social or peer influences resulted in long-term dominance of sedentary behavior and a decline of physically active populations. Social interactions between sedentary and moderately active populations were the most important social parameter that influenced low active populations to become and remain physically active. On the other hand, social interactions encouraging moderately active individuals to become sedentary drove exercise persistence to extinction. Communities should focus on increasing social interactions between sedentary and moderately active individuals to draw sedentary populations to become more active. Additionally, reducing opportunities for moderately active individuals to engage with sedentary individuals through sedentary social activities should be addressed.

data sharing and is dependent on the type of request. Requests can be made at paul.evangelista@westpoint.edu. The code used in this study is available at the link.

**Funding:** DMT was funded by the National Institutes of Health Grant Number U54TR004279. JOH and DS were funded by the National Institutes of Health Grant Number P30DK056336 CKM was funded by the National Institutes of Health awards: P30DK072476, U54 GM104940 ADB was funded by the National Institutes of Health awards, OT2OD026552, UG1HD107711, U19AI14418, UO1CA196406 The funders had no role in study design, data collection and analysis, decision to publish, or preparation of the manuscript.

**Competing interests:** The authors have declared that no competing interests exist.

## Introduction

The new physical activity guidelines published in 2018 provide physical activity recommendations based on scientific evidence connecting physical activity and health [1, 2]. Unfortunately, national trends for meeting physical activity guidelines show little improvement with only 23.2% of adults aged 18 and over meeting the recommendations [3–6]. Community-based physical activity interventions are promoted to facilitate population-wide increases in physical activity and decreases in sedentary behavior [6]. However, community-based interventions to increase physical activity have demonstrated modest effectiveness for altering health behaviors, but systematic reviews and meta-analyses of community-based interventions have also shown substantial variability in the effectiveness of these interventions across multiple age groups, delivery sites, and disease conditions [7–11]. Factors related to personal contact, social support, and social dynamics are emerging as promising candidates to at least partially explain the effectiveness of community- and other group-based interventions [7, 11].

Since social dynamics and influence have been identified as positive factors toward increasing and sustaining physical activity [7, 11–14], mathematical models like the Kermack-McKendrick equations [15], that include social influences population-wide activity trends can help address how to leverage social influence for successful long-term outcomes. Here we use a differential equation model based on the ideas of Kermack and McKendrick [15–20] and the unique data available at the United States Military Academy to estimate parameters and thresholds required to draw and sustain individuals toward improved physical activity habits. The models and conclusions can be used to inform public health community interventions on the conditions that promote wide-spread positive health behavior change.

## Methods

### Study design

A system of differential equations was developed to simulate social influences on exercise persistence in a population. The differential equation model is not data driven and uses a "bottom-up approach" that relies on insights and mechanisms to define interconnections. To identify the known insights that would contribute to the model, we relied on the literature and supporting physical performance data from the United States Military Academy. Since parameters that measure the social influence are extremely difficult to obtain, we choose a range of theoretical values based on the United States Military Academy physical performance data. The United States Military Academy data was collected in the years 2019 and 2020, with the modeling and statistical analysis performed in 2020.

### Army physical fitness test database

The Army Physical Fitness Test (APFT) was the physical fitness assessment of record for the United States Army from 1982 to 2022 [21]. The APFT was recently replaced by the Army Combat Fitness Test in 2022 [22]. The APFT scores, gender, and age were extracted for military officers (civilians were not tested) in 15 USMA academic departments (**Table 1**). The APFT was a three-event assessment consisting of the total number of push-ups performed within two minutes, the total number of sit-ups performed within two minutes, and a timed two-mile run. The APFT was administered to all US Army soldiers every six months.

This study was reviewed and approved by the United States Military Academy Institutional Review Board #19–104. Consent was not required since the study was determined exempt as a secondary analysis of existing de-identified data.

**Table 1. Summary characteristics of participants by department/unit.** Total numbers, age (years), and performance on the Army Physical Fitness Test (APFT) 2-mile run event (sec) are presented.

| Academic Unit (N) | Age (years) | Run Time (seconds) |
|---|---|---|
| AB (32) | 36.97±5.53 | 886.66±200.28 |
| AF (25) | 36.92±5.69 | 789.36±306.17 |
| AH (36) | 36.53±4.91 | 841.33±97.09 |
| AJ (22) | 36.68±5.52 | 849.36±109.42 |
| F (21) | 40.00±5.97 | 993.19±96.63 |
| I (23) | 39.17±5.92 | 959.65±339.40 |
| N (20) | 41.60±5.16 | 976.10±253.56 |
| P (22) | 38.50±7.17 | 899.68±101.49 |
| Q (30) | 39.83±5.55 | 1007.20±406.28 |
| S (27) | 38.67±5.92 | 938.67±86.16 |
| T (24) | 39.25±6.39 | 936.46±208.49 |
| V (9) | 43.11±4.96 | 1007.33±138.68 |
| W (42) | 35.86±4.38 | 908.62±277.74 |
| Y (17) | 35.88±5.89 | 769.41±208.41 |
| Z (22) | 37.59±6.12 | 909.18±115.16 |

## Mathematical model development

Mathematical model development requires transformation of insights and data from to a mathematical formulation followed by solving or simulating the mathematical formulation. After solving/simulating interpretation of the simulations are needed to nest the findings back to the real-world scenario[23, 24]. The description of our model development follows the "math modeling triangle" process described in [23].

**State variable definitions.** The U.S. Department of Health and Human Service's 2018 Physical Activity Guidelines Advisory Report [1] recommends adults participate in 150 minutes per week of moderate-intensity aerobic activity, which equates to 30 minutes per day of moderate-intensity aerobic activity for five days a week. Based on these recommendations, we classified the sedentary population as those who exercise less than 150 minutes per week, the moderately active population as those who exercise between 150-to-300 minutes per week, and the extremely active population as those who exercise more than 300 minutes per week. This classification leads to the model state variable definitions:

$S(t)$ = the number of sedentary individuals in the population on week $t$

$E_1(t)$ = the number of moderately active individuals in the population on week $t$

$E_2(t)$ = the number of extremely active individuals in the population on week $t$

**Description of the flow between model state variables.** The flow between the state variable defined compartments follow the susceptible infected recovered (SIR) infectious disease modeling approach [25] introduced by Kermack and McKendrick [15]. In our model, we allow the flow between state variables to occur spontaneously or be influenced by social interactions. **Fig 1** describes the flow between compartments while **Table 2** describes the variables associated with each flow. We modelled the flow changes due to social interactions using the law of mass action and spontaneous mobility using linear terms.

The resulting mathematical model forms a system of three differential equations, each equation describes the rate of change of state variable at any given time. Complete mathematical details appear in the **S1 Appendix**.

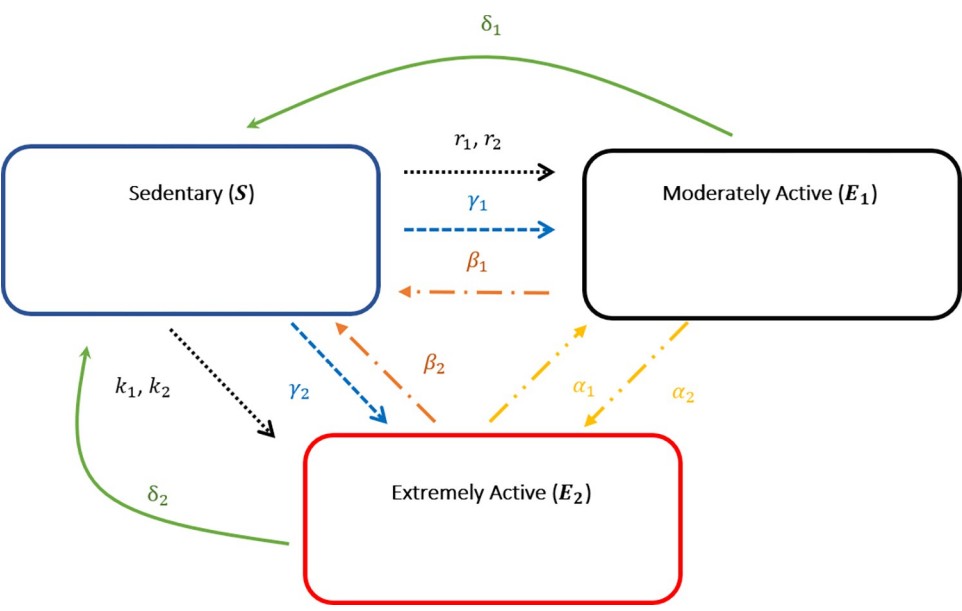

**Fig 1. Flow diagram describing model transitions between state variable populations.**

**Calculation of the basic reproduction rate, $R_0$, and physical activity persistence.** The basic reproduction number $R_0$ has the property that if $R_0<1$, the physically active populations decreases to zero; whereas if $R_0>1$, the physically active populations plateau at a non-zero value or "physical activity persists." Because we are interested in persistence of physical activity, a basic reproduction number $R_0>1$ is desirable. The next generation matrix method [26] was used to compute $R_0$ (details appear in the **S1 Appendix**).

**Model parameters, sensitivity, and simulation.** A scenario was simulated by setting parameters in the different regions delineated by $R_0<1$ and $R_0>1$. The model was also numerically simulated with one individual parameter varied while holding the other parameters fixed. Model simulations used the built-in differential equation solver in the programming platform MATLAB (MathWorks Inc., Nadick, MA 2020).

## Web-based application

A user interfaced application to simulate the system of differential equations was written in RShiny (**RShiny** 2020) and uploaded to the web (https://diana-thomas.shinyapps.io/Exercise/).

## Results

### USMA army physical fitness test

There were N = 372 officers who took the Army Physical Fitness Test for record (**Table 1**). A box plot with the two-mile run time in seconds by academic department (**Fig 2**) revealed that nearly all data points for the slowest times occurred in pairs or small groups. These findings suggest that many of the slower participants, who were unlikely to have identical aerobic (running) fitness levels, made the social decision to run, jog, or walk the two-mile event together.

**Table 2. Description of all model state variables and assumptions related to the state variable and terms of the state equation.**

| State Variable | Assumptions related to the State Variable | Mathematical Formulation of terms |
|---|---|---|
| Sedentary population $S(t)$ | A fraction of sedentary individuals become moderately active exercisers, $E_1$. The rate of transition is dependent on<br>• Contact with moderately active individuals<br>• Contact with extremely active individuals<br>• Spontaneous transition, unrelated to social contact.<br>A fraction of sedentary individuals become extremely active exercisers $E_2$. The rate of transition is dependent on<br>• Contact with moderately active individuals<br>• Contact with extremely active individuals<br>• Spontaneous transition, unrelated to social contact.<br>A fraction of the moderately active exercisers $E_1$ individuals transition back to sedentary compartment as result of social interaction with $S$.<br>A fraction of the extremely active exercisers $E_2$ individuals transition back to sedentary compartment as result of social interaction with $S$.<br>A fraction of the moderately active exercisers $E_1$ individuals spontaneously transition back to sedentary compartment.<br>A fraction of the extremely active exercisers $E_2$ individuals spontaneously transition back to sedentary compartment.<br>Number of sedentary individuals at week 0 is a positive value $S_0$. | $-r_1 \frac{E_1 S}{N}$<br><br>$-r_2 \frac{SE_2}{N}$<br><br>$-\gamma_1 S$<br><br>$-k_1 \frac{SE_1}{N}$<br><br>$-k_2 \frac{SE_2}{N}$<br><br>$-\gamma_2 S$<br><br>$+\beta_1 \frac{E_1 S}{N}$<br><br>$+\beta_2 \frac{SE_2}{N}$<br><br>$+\delta_1 E_1$<br><br>$+\delta_2 E_2$<br><br>$S_0$ |
| Moderately active exercisers $E_1(t)$ | A fraction of sedentary individuals transition to moderately active exercisers $E_1$. The rate of transition is dependent on<br>• Contact with moderately active individuals<br>• Contact with extremely active individuals<br>• Spontaneous transition, unrelated to social contact.<br>A fraction of the moderately active exercisers $E_1$ individuals transition back to sedentary compartment as result of social interaction with $S$.<br>A fraction of the moderately active exercisers $E_1$ individuals transition to extremely active exercisers $E_2$ compartment as result of social interaction with $E_2$.<br>A fraction of the extremely active exercisers $E_2$ individuals transition back to moderately active exercisers $E_1$ compartment as result of social interaction with $E_1$.<br>A fraction of the moderately active exercisers $E_1$ individuals spontaneously transition back to sedentary compartment.<br>Number of moderately active exercisers $E_1$ individuals at week 0. | $-r_1 \frac{E_1 S}{N}$<br><br>$-r_2 \frac{SE_2}{N}$<br><br>$-\gamma_1 S$<br><br>$-\beta_1 \frac{E_1 S}{N}$<br><br>$-\alpha_2 E_1$<br><br>$+\alpha_1 E_2$<br><br>$-\delta_1 E_1$<br><br>$E_1^0$ |
| Extremely active exercisers $E_2(t)$ | A fraction of sedentary individuals transition to extremely active exercisers $E_2$. The rate of transition is dependent on<br>• Contact with moderately active individuals<br>• Contact with extremely active individuals<br>• Spontaneous transition, unrelated to social contact.<br>A fraction of the extremely active exercisers $E_2$ individuals transition back to sedentary compartment as result of social interaction with $S$.<br>A fraction of the moderately active exercisers $E_1$ individuals transition to extremely active exercisers $E_2$ compartment as result of social interaction with $E_2$.<br>A fraction of the extremely active exercisers $E_2$ individuals transition back to moderately active exercisers $E_1$ compartment as result of social interaction with $E_1$.<br>A fraction of the extremely active exercisers $E_2$ individuals spontaneously transition back to sedentary compartment.<br>Number of extremely active exercisers $E_2$ individuals at week 0. | $+k_1 \frac{SE_1}{N}$<br><br>$+k_2 \frac{SE_2}{N}$<br><br>$+\gamma_2 S$<br><br>$-\beta_2 \frac{E_2 S}{N}$<br><br>$+\alpha_2 E_1$<br><br>$-\alpha_1 E_2$<br><br>$+\delta_2 E_2$<br><br>$E_2^0$ |

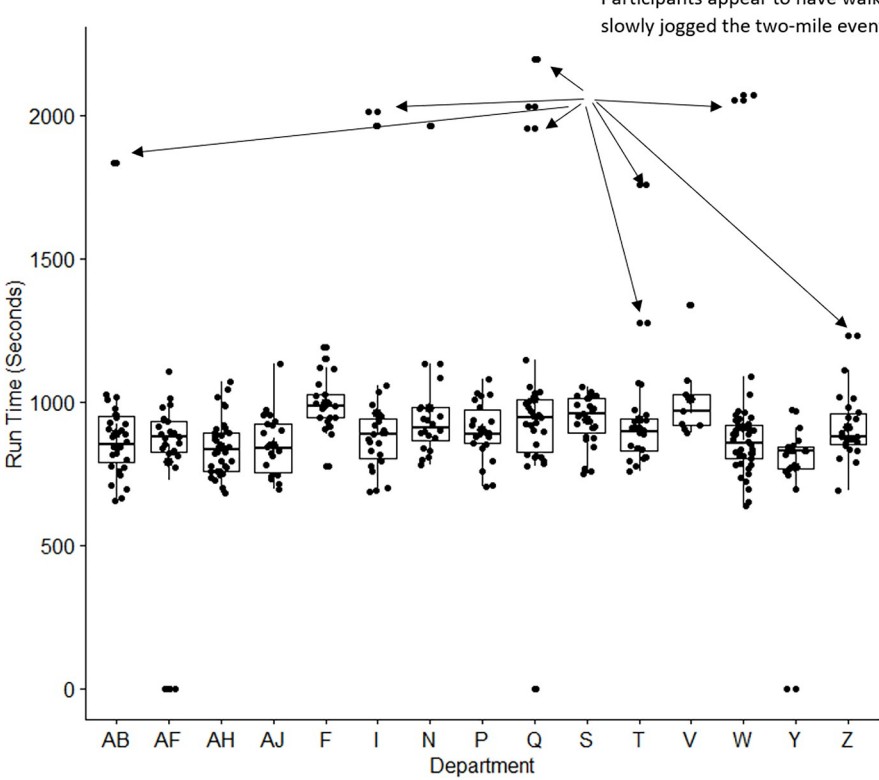

**Fig 2. A box plot with the distribution of times on the two-mile run event by department.** The pairs of individuals achieving the exact same run-time suggests the pair were walking or slowly jogging together.

### Mathematical model predicting exercise persistence in a community

The mathematical model formulated using the flow chart in **Fig 1** and the assumptions detailed in **Table 2** is given by the following system of equations:

$$S'(t) = \beta_1 \frac{E_1(t)S(t)}{N} - r_1 \frac{S(t)E_1(t)}{N} - r_2 \frac{S(t)E_2(t)}{N} - \gamma_1 S(t) + \beta_2 \frac{S(t)E_2(t)}{N} - k_1 \frac{S(t)E_1(t)}{N(t)}$$
$$- k_2 \frac{S(t)E_2(t)}{N} - \gamma_2 S(t) + \delta_1 E_1(t) + \delta_2 E_2(t)$$

$$E_1'(t) = r_1 \frac{S(t)E_1(t)}{N} + r_2 \frac{S(t)E_2(t)}{N} + \gamma_1 S(t) - \beta_1 \frac{E_1(t)S(t)}{N} - \alpha_2 E_1(t) + \alpha_1 E_2(t) - \delta_1 E_1(t)$$

$$E_2'(t) = k_1 \frac{S(t)E_1(t)}{N} + k_2 \frac{S(t)E_2(t)}{N} + \gamma_2 S(t) - \alpha_1 E_2(t) + \alpha_2 E_1(t) - \beta_2 \frac{E_2(t)S(t)}{N} - \delta_2 E_2(t)$$

with initial conditions $S(0) = S_0 \geq 0$, $E_1(0) = E_1^0 \geq 0$ and $E_2(0) = E_2^0 \geq 0$.

### Conditions where physically active populations will persist

The calculation of $R_0$ resulted in:

$$R_0 = \frac{B + \sqrt{B^2 - 4AC}}{2A}$$

where $A$, $B$, and $C$ are combinations of the parameters $k_{1,2}$, $r_{1,2}$, $\alpha_{1,2}$, $\beta_{1,2}$, $\delta_{1,2}$, $S_0$, $N$ given by:

$$A = N^2(\alpha_1\delta_1 + \delta_2(\alpha_2 + \delta_1)) + NS_0(\beta_1(\alpha_1 + \delta_2) + \beta_2(\alpha_2 + \delta_1)) + S_0^2\beta_1\beta_2$$
$$B = S_0[N(\alpha_1k_1 + \delta_1k_2) + N(\alpha_2(k_2 + r_2) + r_1(\alpha_1 + \delta_2)) + S_0(\beta_1k_2 + \beta_2r_1)]$$
$$C = (r_1k_2 - r_2k_1)S_0^2$$

Setting $R_0 > 1$ to find parameter ranges under which exercise habits persist in a population was mathematically intractable. Therefore, we considered special cases.

**Case 1. There are only spontaneous movements between the compartments.** In this case, there is no social influence for either exercising or becoming sedentary. An example of this is when someone decides to exercise because their physician guided them to (not necessarily that their social network influenced their behavior). Another example is that someone may become sedentary due to injury. This scenario allows for interactions between active individuals (moderate and extreme) and allows for active individuals to spontaneously decrease their daily exercise habits. Substituting this into the formula for $R_0$ results in $R_0 = 0 < 1$. Therefore, in this case, the exercising population is driven to extinction, and the population becomes entirely sedentary.

**Case 2. There are no social interactions between sedentary and extremely active individuals and no spontaneous movement from active compartments to sedentary.** This case assumed that there are no interactions between sedentary individuals and extremely active individuals. As an example, consider the scenario where an extremely active individual chooses to spend their lunch break at the office gym while a sedentary individual chooses to spend the lunch break at the office break room, limiting the social interactions between them. To simplify the calculations further, we set the spontaneous terms that describe active individuals moving down to sedentary compartment to zero ($r_2 = k_2 = \beta_2 = \gamma_1 = \gamma_2 = \delta_1 = \delta_2 = 0$). The threshold parameter simplified to the following expression:

$$R_0 = \frac{k_1 + r_1}{\beta_1}$$

For $R_0 > 1$, the sum of $k_1$ and $r_1$ must be higher than $\beta_1$. A simulation of this case reveals that sedentary behavior becomes extinct over time with the population transitioning to a plateau consisting of exercisers (moderate and extreme) (Fig 3A). On the other hand, if the sum of $k_1$

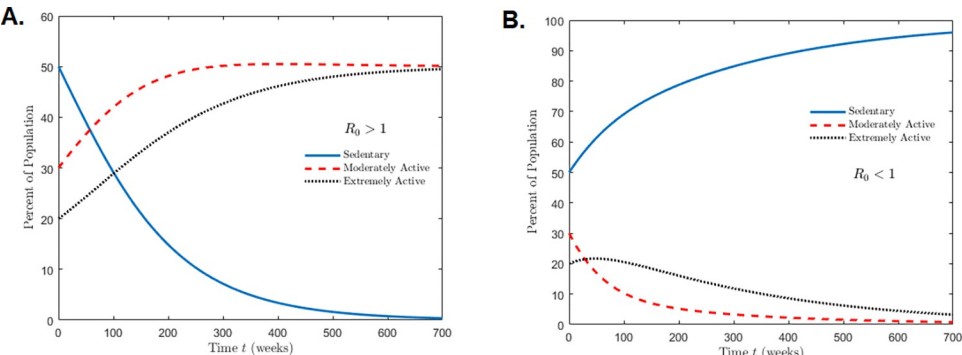

**Fig 3.** Figure simulates the different outcomes when $R_0 > 1$ (**A**. persistence of exercise) and $R_0 < 1$ (**B**. persistence of sedentary behavior). Initial conditions in both simulations were set as 50% of the population sedentary, 30% moderately active and 20% extremely active. In both simulations, no interactions occur between sedentary and extremely active individuals ($k_2 = r_2 = \beta_2 = 0$). For both **A** and **B**, the social interaction parameters that draw sedentary individuals to exercise were set as $k_1 = 0.002$, $r_1 = 0.015$ while interactions between active compartments are set to $\alpha_1 = \alpha_2 = 0.005$. Interactions between sedentary and moderately active individuals that lead to a decrease in physical activity is set to $\beta_1 = 0.0022$. In **B**, the social parameter that draws moderately active individuals to the sedentary population was increased to $\beta_1 = 0.035$.

and $r_1$ are lower than $\beta_1$, the exercising populations become extinct and the sedentary population persists (**Fig 3B**).

### Case 3. There are no social interactions between sedentary and extremely active individuals and there is no spontaneous movement from extremely active to sedentary

This case assumed that sedentary and extremely active groups do not interact with one another and that extremely active individuals do not spontaneously transition to the sedentary compartment. For example, extremely active individuals may be involved in social activities that include a high dose of physical activity while sedentary individuals may have hobbies/ social interests with moderate or low physical activity rate, limiting their social interactions. Additionally, it is unlikely that an extremely active individual will completely become sedentary because of an injury: instead of a two mile run, an extremely active individual may opt for a two mile walk; if badly injured, an extremely active individual may opt to do a light strength routine to maintain activity.In this case,

$R_0$ simplifies to:

$$R_0 = \frac{(k_1 + r_1)\frac{S_0}{N}}{\beta_1 \frac{S_0}{N} + \delta_1}$$

Setting $R_0 > 1$ yields

$$(k_1 + r_1)p > \beta_1 p + \delta_1$$

where $p$ is the percent of the population that is sedentary. Dividing through by $p$ results in:

$$k_1 + r_1 > \beta_1 + \frac{\delta_1}{p}$$

For this inequality to hold, $k_1$ and $r_1$, which are parameters associated with the social draw from the sedentary population to exercise, must dominate over $\beta_1$ and $\frac{\delta_1}{P}$, which are parameters associated with recidivism or can be thought of as "drop out" parameters.

### Sensitivity analysis

The initial conditions were set to 33.3% sedentary individuals, 33.3% moderately active individuals, and 33.3% extremely active individuals. All parameters were fixed at 0.001 while one parameter was varied.

Increasing social interactions with moderately active and sedentary individuals resulted in movement from sedentary to moderately active categories. Spontaneous transition to moderately active also resulted in an increased population of active individuals. A reverse effect was observed when increased social interaction drew exercisers to sedentary behavior. Spontaneous reductions in exercisers also resulted in reduced exercising populations. Social interactions between moderately active individuals and extremely active individuals did not impact the sedentary population.

Increasing parameters associated with spontaneous movement from sedentary to moderately active and extremely active groups led to a 20% decrease in the sedentary population plateau. Increasing the social interaction terms that drive movements from sedentary to active groups led to 13% to 16% decrease in the sedentary population plateau. The spontaneous transition terms increased the sedentary individual plateau by 28% while the social interaction recidivism terms increased the plateau by 21%.

## Web-based application

The web-based app available at https://diana-thomas.shinyapps.io/Exercise/ allows users to simulate the model after entering baseline percentages of sedentary, moderately active, and extremely active populations, as well as social parameter values, recidivism rates, and spontaneous increases in physical activity.

## Discussion

We developed a system of ordinary differential equations that reflect the impact of social interactions between sedentary and active populations on population trends. The model generated a threshold value, $R_0$, which when greater than 1, identifies persistence in exercising. This threshold yields new insights into how to design successful future community interventions and exercise studies that develop persistent exercise habits in a population.

### Social interaction is critical for habitual exercise to persist in a population

When the social influence to motivate the sedentary population to become moderately active individuals was set to zero, the value of $R_0$ was never greater than 1. This means that, over time, an organization without social influence relating to exercise will likely become entirely sedentary.

When present, social interactions regarding physical activity can have different impacts on the individuals' long-term trends. Sensitivity analyses demonstrate that a sedentary individual will more likely transition to the extremely active group when interacting with extremely active individuals and to moderately active group when interacting with moderately active individuals.

### Interventions and strong cultures prevent return to sedetary behaivors

The model indicates that recidivism to sedentary behavior by moderately active individuals is responsible for driving the population to become sedentary. Thus, military bases and units should design methods to reduce drop-out of the moderately active members and limit opportunities to engage in socially sedentary behaviors. It is important to recognize that one factor driving increased sedentariness is the social influence of military members who are sedentary. Thus, it is important to better understand and address factors that contribute to military members engaging in sedentary behavior. These factors could be injury, change in military rank or job responsibilities, familial changes (e.g., the birth of a child, marriage, or divorce), or increased workloads that result in a corresponding lack of time to exercise. Critically, the factors that increase sedentary behavior must be addressed non-punitively without singling-out or 'blaming' military members who experience sedentary behavior. Resources would be needed to ensure that moderately active military peers are able to engage with these individuals to pull them back into a more active lifestyle rather than the more sedentary peer affecting more active peers to become more sedentary. Further work is needed to determine how to do so.

Removing barriers to activity may be one path that reduces spontaneous recidivism. One study found that completers of an exercise intervention were more likely to overcome barriers than counterparts who dropped out [27]. Increased barriers to exercise are associated to lack of exercise adherence [28–30]. Therefore, a method to decrease recidivism could be to reduce burdens by increasing gym access and flexible exercise schedules, particularly for military members who experience events or life changes associated with increased sedentary behavior.

Other possibilities can be drawn from studies with low dropout rates. A recent study by Lee et al. [31] achieved substantially higher participant retention rates by seeking formative

## Extremely active individuals are unlikely to change their exercise habits due to social influence and have little social influence on the sedentary population

The model predictions suggest that extremely active individuals play little role in the overall population dynamics. The model also suggested that interactions with the extreme exercise population did little to change the overall population dynamics. We did not find literature on how extreme exercisers influence peers, however, studies on exercise dropouts found that exercise completers had high exercise self-efficacy [32]. Taken together, this suggests that intervention efforts be placed on moderately active and sedentary groups.

## Sedentary individuals can socially draw the moderate-active population toward sedentary behavior

While moderately active individuals can draw sedentary individuals to be more active, the opposite can also happen. Our analysis of the APFT found that officers who walked or slowly jogged during the two-mile run did so in pairs or groups. When the model included high social influence by the sedentary population, the exercising population reduced to extinction. This finding is supported in the literature, which shows social interactions with sedentary populations can draw moderately active individuals to become sedentary [33]. For example, a social network analysis of 557 children found that peer exposure was positively related to drawing individuals towards both healthful behaviors and unhealthful behaviors [33]. There is also evidence that sedentary leisure time activities such as video game playing are typically performed socially [34].

Most research performed in this space has been with adolescents and little has been investigated with adult populations. While more research is needed on how social interactions may draw individuals to become sedentary, the military is particularly well-positioned to maintain the activity levels of its members due to the influence that the military has on the availability and location of physical resources (e.g., gyms, childcare, etc.), as well as the ability to help soldiers who are at risk for sedentary behavior maintain or increase their activity levels via both social support and physical resources.

## Involving participants in intervention design

Although our model did not include influence with participant engagement during intervention design, a review of the published literature revealed that participant engagement early in the intervention planning process can help reduce participant dropout [35]. In addition, a recent study [31] found that engaging adolescent participants to develop and implement behavioral obesity interventions for adolescent obesity can improve the participant acceptability and effectiveness.

Another large-scale effort to include participant engagement is the ongoing All of Us (AoU) Research Study [36]. The AoU Research Study will enroll one million participants representing the full diversity of the United States who agree to share their electronic health record (EHR) and provide one time blood, urine, and/or saliva samples. The AoU program is designed around the basic philosophy of: "Participants in the study are *partners* in the research." For effective implementation of this philosophy the program includes ongoing

communication between participants and the study team, in large part via the AoU web/app portal [37]. Return of value to participants includes return of research results [36].

## Recommendations to maintain or increase physical activity

Based on the model findings and the literature, we recommend that community-based physical activity interventions develop social activities engineered to increase positive interaction between sedentary and moderately active individuals.

○ Design moderately active social events such as volleyball games

○ Conduct informational meetings while walking

○ Hold scavenger hunts or obstacle course type moderately active engagements similar to those designed for USMA new instructor orientation

- Reduce participant physical activity burden

  ○ Provide easy access to gyms, recreational facilities, and/or equipment

  ○ Develop exercise programs that have convenient timings and are feasible for busy schedules

- Reduce recidivism by moderately active individuals by increasing participant investment in the process.

  ○ Develop mechanisms for continually and deliberately obtaining participant feedback on challenges and burdens that impact retention

  ○ Involve participants in interventions/group exercise design

## Study limitations

Our study has several limitations. We derived insights from a population that is in a closed and active environment and who may be even more active than the average other military environment. USMA has a culture of being physically active. Its ubiquitous social engagements around fitness may not be reflective of other military bases or environments. Despite this limitation, USMA provides a mixture of the different active populations in a military environment that give insight into social interactions and their influence on lifestyle habits. Moreover, the differential equation model was not specifically tailored for USMA only, as its parameters can be set to reflect characteristics of different environments.

Our model was also not validated on data. To do so would require a closed environment with conditions to test model predictions on which would be unfeasible; however, models such as the one we developed are more useful for study design and intervention development than prediction.

Finally, we note that most exercise interventions studies do not follow the participants who dropped out of the study. In several investigations that have studied participants who dropped out [38], researchers identified factors that differentiate completers from dropouts. More systematic research needs to be performed on the dropout cohort to better understand drop out motivation and factors that influence retention.

## Conclusion

Here. we have established criteria on social influences that lead to exercise persistence. Specifically, we found that the social interaction between moderately active individuals with the

sedentary population should be engineered to draw the sedentary individuals into more active behaviors. A second criteria identified that strategies are needed to control dropouts from moderately active individuals. Finally, we found that discouraging socially acceptable sedentary behavior will reduce drop out of moderately active individuals. These findings can be used to design and implement community based exercise programs that lead to long-term exercise persistence in the community.

## Supporting information

**S1 Appendix. A detailed self-contained mathematical description of the model appears in the supporting information.**
(DOCX)

## Acknowledgments

DMT and ES conceived this study. EM performed all simulations and differential equation analysis. LE and HDM secured the APFT physical activity data. JOH, DS, CKM, AB, and NG reviewed and implemented the literature for the systems models. DMT, DS, LE, HDM, and EM wrote initial drafts of the manuscript. All authors reviewed and approved the manuscript. The views expressed in this work are those of the authors and do not reflect the official policy or position of the United States Military Academy, Department of the Army, or the Department of Defense. The results of the study are presented clearly, honestly, and without fabrication, falsification, or inappropriate data manipulation.

## Author Contributions

**Conceptualization:** Everett S. Spain, Corby K. Martin, James O. Hill, Diana M. Thomas.

**Formal analysis:** Ensela Mema, Lee A. Evans, Diana M. Thomas.

**Investigation:** Everett S. Spain, Diana M. Thomas.

**Methodology:** Ensela Mema, Everett S. Spain, Nicholas H. Gist, Diana M. Thomas.

**Project administration:** Howard D. McInvale.

**Supervision:** Corby K. Martin, Diana M. Thomas.

**Visualization:** Ensela Mema, Lee A. Evans.

**Writing – original draft:** Everett S. Spain, James O. Hill, R. Drew Sayer, Lee A. Evans, Nicholas H. Gist, Alexander D. Borowsky, Diana M. Thomas.

**Writing – review & editing:** Ensela Mema, Everett S. Spain, Corby K. Martin, James O. Hill, R. Drew Sayer, Howard D. McInvale, Lee A. Evans, Nicholas H. Gist, Alexander D. Borowsky, Diana M. Thomas.

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
