## [Decision Letter · Decision Letter 0]

1 Aug 2022

PONE-D-22-17305A Systems Model Describing the Social Influence on Physical Activity in a CommunityPLOS ONE

Dear Dr. Thomas,

Thank you for submitting your manuscript to PLOS ONE. After careful consideration, we feel that it has merit but does not fully meet PLOS ONE’s publication criteria as it currently stands. Therefore, we invite you to submit a revised version of the manuscript that addresses the points raised during the review process.

ACADEMIC EDITOR: Dear Author, Based on the reviewers' comments your manuscript still requires some corrections. Please attends to all the reviewers comments and make the necessary changes. The decision of this manuscript is justified based on PLOS ONE’s publication criteria and not on its novelty or perceived impact.

We look forward to receiving your revised manuscript.

Kind regards,

Zulkarnain Jaafar

Academic Editor

PLOS ONE

Journal Requirements:

2. Please note that PLOS ONE has specific guidelines on code sharing for submissions in which author-generated code underpins the findings in the manuscript. In these cases, all author-generated code must be made available without restrictions upon publication of the work. Please review our guidelines at https://journals.plos.org/plosone/s/materials-and-software-sharing#loc-sharing-code and ensure that your code is shared in a way that follows best practice and facilitates reproducibility and reuse. Any software must comply with the Open Source Definition.

NO authors have competing interests

Enter: The authors have declared that no competing interests exist.

Reviewers' comments:

Reviewer's Responses to Questions

**Comments to the Author**

1. Is the manuscript technically sound, and do the data support the conclusions?

Reviewer #1: Yes

Reviewer #2: Partly

2. Has the statistical analysis been performed appropriately and rigorously? 

Reviewer #1: Yes

Reviewer #2: N/A

3. Have the authors made all data underlying the findings in their manuscript fully available?

Reviewer #1: Yes

Reviewer #2: Yes

4. Is the manuscript presented in an intelligible fashion and written in standard English?

Reviewer #1: Yes

Reviewer #2: Yes

5. Review Comments to the Author

Reviewer #1: The report presents the outcome of a study on the social influence on physical activity in a community. The contribution of the report to the body of knowledge is significant and novel. Also, the aim and objectives of the study are within the scope of your journal. However, the present form of the report needs revision. The author should consider the following points:

Q1. Page 8 of 39, abstract, it was written:

Despite well-documented health benefits from exercise, national trends in achieving the recommended minutes of physical activity guidelines have not changed.

Comment: This sounds like a harsh conclusion without any basis. Revise the sentence and make it to accurately presents the significance of the study.

Q2. Page 8 of 39, result section of the abstract, line 46, it was written:

A web-based app to simulate model the systems model is available at https://diana-thomas.shinyapps.io/Exercise/.

Comment: Delete because it seems irrelevant.

Q3. Line 44, it was written, "...moderately activey that drew moderately active..."

Comment: What do you mean by that? Seems faulty. Revise.

Q4. Line 69, introduction, it was written:

We apply a Kermack-McKendrick model to investigate how social interactions influence the prevalence of active individuals in a population.

Comment: Authors have violated the significance of the introduction section. The statement above was actually wrongly used to present the aim of the study.

Q5. There is need to restructure the entire introduction section such that you would be able to

a) describe the rationale for undertaking the study.

b) explain how the research makes an important contribution to the field of advanced knowledge.

c) state the research question clearly.

d) explain the theoretical framework that the study is based on.

d) provide a background of the problem or issue that your research aims to understand or resolve, citing studies to support your arguments.

e) summarize the current state of knowledge on the topic, citing studies as appropriate.

Don't review all studies that have ever been published on the topic.

Comment: Revise the introduction section.

Q6. Merge the introduction to form just two paragraphs.

Q7. The introduction was not structured enough to announce the existing published facts on social influences on exercise persistence in a population.

Comment: How is it possible to address this shortcoming?

Q8. Line 95, it was written:

The APFT was administered to all US Army soldiers every six months.

Comment: For how many years? Update.

Q9. Line 128, it was written:

We simulated a scenario from insights obtained from the survey responses and we set parameters in the different regions delineated by 0 < 1 and 0 > 1.

Comment: The sentence above is faulty because of wrong useage of the pronoun, "we" in a scientific report like this.

Q10. Line 147, the difference equation was presented as if the authors followed the first principle of derivation. That's not true.

Comment: Update with at least a published source of the related model. However, some sentences are needed to present the development of the components of the differential equation.

Q11. Line 216, it was merely written, "DISCUSSION"

Comment: Discussion of what?

Q12. The conclusion section needs revision.

Comment: A sentence is needed to re-present the aim of the study elaborately. Then remark that the major objectives had been established. Start the conclusion section with a fact on the achievement of the research aim before stating conclusive statements. Revise the title to present the aim concisely. The aim of the title should reflect in the abstract, and at the beginning of the conclusion.

Revise the conclusion section to provide conclusive statements on the research questions posed at the end of the introduction.

Q13. Update the report with a published fact on mathematical modeling. The author may consider, “According to Ref A et al. (2022), mathematical modeling involves analyzing real-world situations using mathematical terminology and involves transforming them into a mathematical form.”

Ref A et al. (2022): Ratio of Momentum Diffusivity to Thermal Diffusivity: Introduction, Meta-analysis, and Scrutinization. Chapman and Hall/CRC. New York, 2022. ISBN-13: 978-1032108520, ISBN-10: 1032108525, ISBN9781003217374

Q14. Back to the title, it was written:

A Systems Model Describing the Social Influence on Physical Activity in a Community

Comment: Is it possible to revise the title to reflect the content of discussion of results? There is need for the title to accurately connect with the discussion of results.

Reviewer #2: 1. Real life situations under which the scenarios given in cases 1-3 (lines 162-193) persist must be included.

2. Estimates of the model parameters must be given or provide citations if values are taken from the literature.

3. How were the parameter values used for Figure 3 chosen? Explain how.

4. Explain clearly the novelty in the model.

6. PLOS authors have the option to publish the peer review history of their article (what does this mean?). If published, this will include your full peer review and any attached files.

Reviewer #1: **Yes: **Dr. ANIMASAUN, Isaac Lare

Reviewer #2: No

---

## [Author Response · Author response to Decision Letter 0]

9 Aug 2022

We thank the reviewers for their reading of our manuscript and the helpful comments. We have listed each reviewer comment below, followed by the response to the comment. For the reviewer’s convenience, the changes to the manuscript based on the comment have been pasted below our response in red.

Reviewer Comments

Reviewer 1

Reviewer 1 Comment 1: Page 8 of 39, abstract, it was written:

Despite well-documented health benefits from exercise, national trends in achieving the recommended minutes of physical activity guidelines have not changed.

Comment: This sounds like a harsh conclusion without any basis. Revise the sentence and make it to accurately presents the significance of the study.

Response to Reviewer 1 Comment 1: The reviewer makes a great point. There is, unfortunately, national evidence in a JAMA open publication derived from both self-reported physical activity and objective accelerometry to show that in fact physical activity trends have declined. The authors of that study and two others that we cite conclude that efforts to increase physical activity and reduce sedentary time are needed in the US.

From the conclusions from our reference number 4 where PAG stands for physical activity guidelines: Du Y, Liu B, Sun Y, Snetselaar LG, Wallace RB, Bao W. Trends in Adherence to the Physical Activity Guidelines for Americans for Aerobic Activity and Time Spent on Sedentary Behavior Among US Adults, 2007 to 2016. JAMA Netw Open. 2019;2(7):e197597. 

“The findings suggest that the adherence rate to the PAG for aerobic activity in US adults has not improved since the release of the first edition in 2008 but that time spent on sedentary behavior has significantly increased over time. Further nationwide efforts appear to be warranted to not only promote physical activity but also reduce sedentary time in the United States.”

We have revised the sentence in the abstract to reflect this evidence and mirror the language of this study:

Background: Despite well-documented health benefits from exercise, a study on national trends in achieving the recommended minutes of physical activity guidelines has not improved since the guidelines were published in 2008. 

Reviewer 1 Comment 2: Page 8 of 39, result section of the abstract, line 46, it was written:

A web-based app to simulate model the systems model is available at https://diana-thomas.shinyapps.io/Exercise/.

Comment: Delete because it seems irrelevant.

Response to Reviewer 1 Comment 2: We have deleted it as suggested.

Reviewer 1 Comment 3: Line 44, it was written, "...moderately activey that drew moderately active..."

Comment: What do you mean by that? Seems faulty. Revise.

Response to Reviewer 1 Comment 2: We have revised this sentence to:

On the other hand, social interactions encouraging moderately active individuals to become sedentary drove exercise persistence to extinction.

Reviewer 1 Comment 4: Line 69, introduction, it was written:

We apply a Kermack-McKendrick model to investigate how social interactions influence the prevalence of active individuals in a population.

Comment: Authors have violated the significance of the introduction section. The statement above was actually wrongly used to present the aim of the study.

Response to Reviewer 1 Comment 4: We now mirror what was stated in the abstract. The change is pasted below:

We apply a Kermack-McKendrick model to dynamically model the social influences on a population’s physical activity and to establish criteria that will generate sustained exercise habits. 

Reviewer 1 Comment 5: There is need to restructure the entire introduction section such that you would be able to

a) describe the rationale for undertaking the study.

b) explain how the research makes an important contribution to the field of advanced knowledge.

c) state the research question clearly.

d) explain the theoretical framework that the study is based on.

d) provide a background of the problem or issue that your research aims to understand or resolve, citing studies to support your arguments.

e) summarize the current state of knowledge on the topic, citing studies as appropriate.

Don't review all studies that have ever been published on the topic.

Comment: Revise the introduction section.

Merge the introduction to form just two paragraphs.

The introduction was not structured enough to announce the existing published facts on social influences on exercise persistence in a population.

Comment: How is it possible to address this shortcoming?

Response to Reviewer 1 Comment 5: 

We have merged the comments that the reviewer had on the introduction as Comment 6. We thank the reviewer for this comment. When re-reading the introduction, we realized that over time the introduction became jumbled with too much information. We have revised the introduction to make the objectives and the point of our study clear. The reviewer had already pointed out that there was no evidence provided that the US population is not meeting the PAG and we have started the introduction with this evidence. We have also explained why we are interested in the social factors that influence increases in population-wide physical activity. We have also reduced the introduction to two paragraphs. The revised introduction is pasted below:

The new physical activity guidelines published in 2018 provide physical activity recommendations based on scientific evidence connecting physical activity and health (1, 2). Unfortunately, national trends for meeting physical activity guidelines show little improvement with only 23.2% of adults aged 18 and over meeting the recommendations (3-6). Community-based physical activity interventions are promoted to facilitate population-wide increases in physical activity and decreases in sedentary behavior (6). However, community-based interventions to increase physical activity have demonstrated modest effectiveness for altering health behaviors, but systematic reviews and meta-analyses of community-based interventions have also shown substantial variability in the effectiveness of these interventions across multiple age groups, delivery sites, and disease conditions(7-11). Factors related to personal contact, social support, and social dynamics are emerging as promising candidates to at least partially explain the effectiveness of community- and other group-based interventions(7, 11).

Since social dynamics and influence have been identified as positive factors toward increasing and sustaining physical activity (7, 11, 16-18), mathematical models like the Kermack-McKendrick equations (19), that include social influences population-wide activity trends can help address how to leverage social influence for successful long-term outcomes. Here we use a differential equation model based on the ideas of Kermack and McKendrick (19-24) and the unique data available at the United States Military Academy to estimate parameters and thresholds required to draw and sustain individuals toward improved physical activity habits. The models and conclusions can be used to inform public health community interventions on the conditions that promote wide-spread positive health behavior change.

Reviewer 1 Comment 6: Line 95, it was written:

The APFT was administered to all US Army soldiers every six months.

Comment: For how many years? Update.

Response to Reviewer 1 Comment 6: We have updated this as suggested by the reviewer. The revision appears below:

Army Physical Fitness Test Database

The Army Physical Fitness Test (APFT) was the physical fitness assessment of record for the United States Army from 1982 to 2022 (25). The APFT was recently replaced by the Army Combat Fitness Test in 2022 (26).

Reviewer 1 Comment 7: Line 128, it was written:

We simulated a scenario from insights obtained from the survey responses and we set parameters in the different regions delineated by 0 < 1 and 0 > 1.

Comment: The sentence above is faulty because of wrong useage of the pronoun, "we" in a scientific report like this.

Response to Reviewer 1 Comment 7: We thank the reviewer for not only the pronoun usage, but also that we are not using a survey. We initially wanted to administer a survey, received IRB approval, administered in one department, and then…COVID happened. Because we are at a federal institution, surveys about faculty/staff behavior during COVID increased and our institution has responded by discouraging any research surveys that would take additional faculty time. We have revised the sentence the reviewer found and we have also removed any lingering references to a survey from the manuscript. The revised sentence is below.

A scenario was simulated by setting parameters in the different regions delineated by R_0<1 and R_0>1. 

Reviewer 1 Comment 8: Line 147, the difference equation was presented as if the authors followed the first principle of derivation. That's not true.

Comment: Update with at least a published source of the related model. However, some sentences are needed to present the development of the components of the differential equation.

Response to Reviewer 1 Comment 8: We have revised the introduction to the model development and cited the literature on SIR models. Because the audience for this study is interdisciplinary we have provided an appendix with a step by step model development was provided with every detail on differential equation model development laid out. The revisions in the manuscript appear below:

The flow between the state variable defined compartments follow the susceptible infected recovered (SIR) infectious disease modeling approach [28] introduced by Kermack and McKendrick [15]. In our model, we allow the flow between state variables to occur spontaneously or be influenced by social interactions. 

Reviewer 1 Comment 9: Line 216, it was merely written, "DISCUSSION"

Comment: Discussion of what?

Response to Reviewer 1 Comment 9: This was the beginning of the discussion section of the manuscript, but the title to the section was at the end of a page making it “hang”. We moved the title to the discussion section to the next page.

Reviewer 1 Comment 10: The conclusion section needs revision.

Comment: A sentence is needed to re-present the aim of the study elaborately. Then remark that the major objectives had been established. Start the conclusion section with a fact on the achievement of the research aim before stating conclusive statements. Revise the title to present the aim concisely. The aim of the title should reflect in the abstract, and at the beginning of the conclusion.

Revise the conclusion section to provide conclusive statements on the research questions posed at the end of the introduction.

Response to Reviewer 1 Comment 10:

The conclusion was revised to connect the aims to the findings. The title was revised to reflect the aim of the study. The revisions to the conclusions are below and the title change follows:

A model that includes social influences on physical activity identified a set of criteria that must be satisfied to ensure that exercise habits increase and persist. A criteria found is that the social interaction between moderately active individuals with the sedentary population should be engineered to draw the sedentary individuals into more active behaviors. A second criteria identified that strategies are needed to control dropouts from moderately active individuals.. Eliminating sedentary environments and discouraging socially acceptable sedentary behavior may reduce potential for socially drawing moderately active individuals to become more sedentary.

A dynamic model of social influences on physical activity used to establish criteria that lead to exercise persistence

Reviewer 1 Comment 11: Update the report with a published fact on mathematical modeling. The author may consider, “According to Ref A et al. (2022), mathematical modeling involves analyzing real-world situations using mathematical terminology and involves transforming them into a mathematical form.”

Ref A et al. (2022): Ratio of Momentum Diffusivity to Thermal Diffusivity: Introduction, Meta-analysis, and Scrutinization. Chapman and Hall/CRC. New York, 2022. ISBN-13: 978-1032108520, ISBN-10: 1032108525, ISBN9781003217374

Response to Reviewer 1 Comment 11: We thank the reviewer for this opportunity to lay out what we call the math modeling triangle: Transform->Solve->Interpret and have laid out an introduction to the model development with this information and a reference. We note that the Appendix has more mathematical detail on model development. We hope that is okay with the reviewer that we cited Frank Giordano who wrote a seminal book on this process.

Mathematical Model Development

 Mathematical model development requires transformation of insights and data from to a mathematical formulation followed by solving or simulating the mathematical formulation. After solving/simulating interpretation of the simulations are needed to nest the findings back to the real-world scenario[28]. The description of our model development follows this “math modeling triangle” [28].

Reviewer 1 Comment 12: Back to the title, it was written:

A Systems Model Describing the Social Influence on Physical Activity in a Community

Comment: Is it possible to revise the title to reflect the content of discussion of results? There is need for the title to accurately connect with the discussion of results.

Response to Reviewer 1 Comment 12: 

We have revised the title to:

A dynamic model of social influences on physical activity used to establish criteria that lead to exercise persistence

Reviewer 2 Comment 1: Real life situations under which the scenarios given in cases 1-3 (lines 162-193) persist must be included.

Response to Reviewer 2 Comment 1: We thank the reviewer for pointing this out. We have revised the manuscript to include a specific real life situations that begin with the “For example,…” statement. The cases have been revised to include the red sentences that appear below:

Case 1: There are only spontaneous movements between the compartments. In this case, there is no social influence for either exercising or becoming sedentary. An example of this is when someone decides to exercise because their physician guided them to (not necessarily that their social network influenced their behavior). Another example is that someone may become sedentary due to injury. This scenario allows for interactions between active individuals (moderate and extreme) and allows for active individuals to spontaneously decrease their daily exercise habits. Substituting this into the formula for R_0 results in R_0=0<1. Therefore, in this case, the exercising population is driven to extinction, and the population becomes entirely sedentary.

Case 2: There are no social interactions between sedentary and extremely active individuals and no spontaneous movement from active compartments to sedentary. This case assumed that there are no interactions between sedentary individuals and extremely active individuals. As an example, consider the scenario where an extremely active individual chooses to spend their lunch break at the office gym while a sedentary individual chooses to spend the lunch break at the office break room, limiting the social interactions between them. To simplify the calculations further, we set the spontaneous terms that describe active individuals moving down to sedentary compartment to zero (r2=k2=β2=γ1 = γ2 = δ1 = δ2 = 0).

 Case 3: There are no social interactions between sedentary and extremely active individuals and there is no spontaneous movement from extremely active to sedentary. This case assumed that sedentary and extremely active groups do not interact with one another and that extremely active individuals do not spontaneously transition to the sedentary compartment. For example, extremely active individuals may be involved in social activities that include a high dose of physical activity while sedentary individuals may have hobbies/ social interests with moderate or low physical activity rate, limiting their social interactions. Additionally, it is unlikely that an extremely active individual will completely become sedentary because of an injury: instead of a two mile run, an extremely active individual may opt for a two mile walk; if badly injured, an extremely active individual may opt to do a light strength routine to maintain activity.

Reviewer 2 Comment 2 & 3: Estimates of the model parameters must be given or provide citations if values are taken from the literature. How were the parameter values used for Figure 3 chosen? Explain how

Response to Reviewer 2 Comments 2& 3: We thank the reviewer for pointing this out and we address both of their comments below.

 We initially wanted to administer a survey and use the data from the survey to estimate parameters. We received IRB approval, administered the survey in one department, and then…COVID happened. Because we are at a federal institution, surveys about faculty/staff behavior during COVID increased and our institution has responded by discouraging any research surveys that would take additional faculty time. We took the responses from the survey administered in one department and used theoretical ranges that were close to the parameters estimated from the survey. We have revised our manuscript to point out that the model parameters used are theoretical and removed any lingering references to a survey from the manuscript. The parameter values used in Figure 3 and subsequent figures of our manuscript are given in the figure legend.

A system of differential equations was developed to simulate social influences on exercise persistence in a population. The differential equation model is not data driven and uses a “bottom-up approach” that relies on insights and mechanisms to define interconnections. To identify the known insights that would contribute to the model, we relied on the literature and supporting physical performance data from the United States Military Academy. Since parameters that measure the social influence are extremely difficult to obtain, we choose a range of theoretical values based on the United States Military Academy physical performance data. The United States Military Academy data was collected in the years 2019 and 2020, with the modeling and statistical analysis performed in 2020.

Reviewer 2 Comment 4: Explain clearly the novelty in the model.

Response to Reviewer 2 Comment 4: We applied a Kermack-McKendrick model, originally used to model infectious disease spread in a closed population to simulate how social influences affect exercise persistence in a population. Since social influence is extremely difficult to measure and such parameters do not exist in literature, the Kermack-McKendrick model allows us to perform sensitivity analysis with a range of theoretical parameters and draw recommendations that will help increase the number of active individuals in a population. We hope that our manuscript inspires community intervention researchers to conduct experiments that measure the social influence on exercise persistence. We have added a sentence that addresses this point in our conclusion section. 

CONCLUSION

A model that includes social influences on physical activity identified a set of criteria that must be satisfied to ensure that exercise habits increase and persist. A criteria found is that the social interaction between moderately active individuals with the sedentary population should be engineered to draw the sedentary individuals into more active behaviors. A second criteria identified that strategies are needed to control dropouts from moderately active individuals. Eliminating sedentary environments and discouraging socially acceptable sedentary behavior may reduce potential for socially drawing moderately active individuals to become more sedentary. We hope that our model and results will inspire future experimental investigations in community intervention research that may provide further insight and lead to improved recommendations.

Editorial Comments:

We have ensured that our formatting matches the sample.

 Please note that PLOS ONE has specific guidelines on code sharing for submissions in which author-generated code underpins the findings in the manuscript. In these cases, all author-generated code must be made available without restrictions upon publication of the work. Please review our guidelines at https://journals.plos.org/plosone/s/materials-and-software-sharing#loc-sharing-code and ensure that your code is shared in a way that follows best practice and facilitates reproducibility and reuse. Any software must comply with the Open Source Definition.

We added a section called DATA AVAILABILITY STATMENT and shared the code using a Dropbox folder link. The statement is below. The data is US Soldier data and we would need to ensure that there are not national security risks when sharing so there is an internal review required.

DATA AVAILABILITY STATEMENT

Data will be made available by request after review by the United States Military Academy Chief Data Officer. The review will determine the risks to personnel for data sharing and is dependent on the type of request. Requests can be made at paul.evangelista@westpoint.edu. The code used in this study is available at the link.

 We note that the grant information you provided in the ‘Funding Information’ and ‘Financial Disclosure’ sections do not match. 

We have checked them and made them match.

We have done so.

 Thank you for stating the following in your Competing Interests section: 

NO authors have competing interests

Enter: The authors have declared that no competing interests exist.

This has been revised as suggested.

---

## [Decision Letter · Decision Letter 1]

22 Aug 2022

PONE-D-22-17305R1A dynamic model of social influences on physical activity used to establish  criteria that lead to exercise persistencePLOS ONE

Dear Dr. Thomas,

Thank you for submitting your manuscript to PLOS ONE. After careful consideration, we feel that it has merit but does not fully meet PLOS ONE’s publication criteria as it currently stands. Therefore, we invite you to submit a revised version of the manuscript that addresses the points raised during the review process.

Please attend to all the reviewers' comments and make the necessary corrections.

The decision of this manuscript is justified based on PLOS ONE’s publication criteria and not on its novelty or perceived impact.

We look forward to receiving your revised manuscript.

Kind regards,

Zulkarnain Jaafar

Academic Editor

PLOS ONE

Journal Requirements:

Reviewers' comments:

Reviewer's Responses to Questions

**Comments to the Author**

1. If the authors have adequately addressed your comments raised in a previous round of review and you feel that this manuscript is now acceptable for publication, you may indicate that here to bypass the “Comments to the Author” section, enter your conflict of interest statement in the “Confidential to Editor” section, and submit your "Accept" recommendation.

Reviewer #1: (No Response)

Reviewer #2: All comments have been addressed

2. Is the manuscript technically sound, and do the data support the conclusions?

Reviewer #1: (No Response)

Reviewer #2: Yes

3. Has the statistical analysis been performed appropriately and rigorously? 

Reviewer #1: (No Response)

Reviewer #2: Yes

4. Have the authors made all data underlying the findings in their manuscript fully available?

Reviewer #1: (No Response)

Reviewer #2: Yes

5. Is the manuscript presented in an intelligible fashion and written in standard English?

Reviewer #1: (No Response)

Reviewer #2: Yes

6. Review Comments to the Author

Reviewer #1: The report presents the outcome of a study on the social influence on physical activity in a community. The contribution of the report to the body of knowledge is significant and novel. Also, the aim and objectives of the study are within the scope of your journal. However, the present form of the report needs revision. The author should consider the following points:

Q1. The title presents something impossible. How is it possible that dynamic model of social has influence on physical activity?

Revise the title or consider:

Social influences on physical activity for establishing criteria leading to exercise persistence

Q2. Line 42, it was written:

Objectives: The objective of this study is to dynamically model social influences on physical activity to establish quantitative criteria that lead to sustained exercise habits.

Comment: Revise the objectives to present the objectives of the research for social influences on physical activity for establishing criteria leading to exercise persistence

Q3. It is now necessary to delete the headings under the abstract like: Background:, Objectives:, Methods:, Results:, and Conclusions:.

Q4. Ensure that the abstract is not up to 250 words.

Q5. The authors did not respond to

Q13. Update the report with a published fact on mathematical modeling. The author may consider, “According to Ref A et al. (2022), mathematical modeling involves analyzing real-world situations using mathematical terminology and involves transforming them into a mathematical form.”

Ref A et al. (2022): Ratio of Momentum Diffusivity to Thermal Diffusivity: Introduction, Meta-analysis, and Scrutinization. Chapman and Hall/CRC. New York, 2022. ISBN-13: 978-1032108520, ISBN-10: 1032108525, ISBN9781003217374

Q6. The revised version is difficult to understand. Remove the first version. Delete the old sentence. You just need to highlight the revised path. Do not underline any sentence.

Q7. Line 417, it was written:

We hope that our model and results will inspire future experimental investigations in community intervention research that may provide further insight and lead to improved recommendations.

One of the most crucial parts of the manuscript is the conclusion. By examining the key ideas in the report, bringing these ideas to a convincing conclusion, and presenting a concluding viewpoint, it makes a lasting effect on the reader.

Comment: The authors should try to give the reader a feeling of closure with a concluding sentence. This is needed to boost the impact of authors claim. A sentence is needed to first re-present the aim of the study elaborately. Then remark that the major objectives had been established. Start the conclusion section with a fact on the achievement of the research aim before stating conclusive statements. Revise the title to present the aim concisely. The aim of the title should reflect in the abstract, and at the beginning of the conclusion. Author's should revise the conclusion section to provide conclusive statements on the research questions posed at the end of the introduction. The conclusion section should be revised.

Reviewer #2: (No Response)

7. PLOS authors have the option to publish the peer review history of their article (what does this mean?). If published, this will include your full peer review and any attached files.

Reviewer #1: **Yes: **Dr. ANIMASAUN, Isaac Lare

Reviewer #2: No

---

## [Author Response · Author response to Decision Letter 1]

23 Aug 2022

Response to Reviewer 1

We thank the reviewer for spending the time to carefully go through the manuscript again. We have responded to each comment and any change made in the manuscript has been copied and pasted in red for the convenience of the reviewer.

Reviewer #1: The report presents the outcome of a study on the social influence on physical activity in a community. The contribution of the report to the body of knowledge is significant and novel. Also, the aim and objectives of the study are within the scope of your journal. However, the present form of the report needs revision. The author should consider the following points:

Comment Number 1: The title presents something impossible. How is it possible that dynamic model of social has influence on physical activity?

Revise the title or consider:

Social influences on physical activity for establishing criteria leading to exercise persistence

Response to Comment Number 1: We were trying to get in that this was a mathematical model, but agree with the reviewer that it turns out to a mouthful of nothing. We liked the title the reviewer suggested and changed our title to the suggested title.

Comment Number 2: Line 42, it was written:

Objectives: The objective of this study is to dynamically model social influences on physical activity to establish quantitative criteria that lead to sustained exercise habits.

Comment: Revise the objectives to present the objectives of the research for social influences on physical activity for establishing criteria leading to exercise persistence

Response to Comment Number 2: We have changed the objective as suggested:

The objective of this study is for establishing criteria for social influences on physical activity for establishing criteria that lead to exercise persistence.

Comment Number 3 & 4: It is now necessary to delete the headings under the abstract like: Background:, Objectives:, Methods:, Results:, and Conclusions:. Ensure that the abstract is not up to 250 words.

Response to Comment Number 3 & 4: We have removed the structured portion of the abstract. The word count is 229 words.

Comment Number 5: The authors did not respond to. Update the report with a published fact on mathematical modeling. The author may consider, “According to Ref A et al. (2022), mathematical modeling involves analyzing real-world situations using mathematical terminology and involves transforming them into a mathematical form.”

Ref A et al. (2022): Ratio of Momentum Diffusivity to Thermal Diffusivity: Introduction, Meta-analysis, and Scrutinization. Chapman and Hall/CRC. New York, 2022. ISBN-13: 978-1032108520, ISBN-10: 1032108525, ISBN9781003217374

Response to Comment Number 5: We did in fact respond to this. We referenced a seminal mathematical modeling book that we use to teach at West Point. The textbook includes the mathematical modeling triangle that starts from transform form a real world problem to solve the mathematical formulation to interpret the solution. However, we have now included the suggested reference as well. The paragraph is below with the suggested reference as Number 24.

 Mathematical model development requires transformation of insights and data from to a mathematical formulation followed by solving or simulating the mathematical formulation. After solving/simulating interpretation of the simulations are needed to nest the findings back to the real-world scenario[23, 24]. The description of our model development follows the “math modeling triangle” process described in [23].

Comment Number 6: The revised version is difficult to understand. Remove the first version. Delete the old sentence. You just need to highlight the revised path. Do not underline any sentence.

Response to Comment Number 6: We understand, however, the PLoS One guidelines indicated we needed to use track changes which is what we have done. We have the clean version and the snippets cut here so that the reviewer can read them. We have also highlighted changes and supplied the highlighted version as supplemental materials for the reviewer only.

Comment Number 7: Line 417, it was written:

We hope that our model and results will inspire future experimental investigations in community intervention research that may provide further insight and lead to improved recommendations.

One of the most crucial parts of the manuscript is the conclusion. By examining the key ideas in the report, bringing these ideas to a convincing conclusion, and presenting a concluding viewpoint, it makes a lasting effect on the reader. The authors should try to give the reader a feeling of closure with a concluding sentence. This is needed to boost the impact of authors claim. A sentence is needed to first re-present the aim of the study elaborately. Then remark that the major objectives had been established. Start the conclusion section with a fact on the achievement of the research aim before stating conclusive statements. Revise the title to present the aim concisely. The aim of the title should reflect in the abstract, and at the beginning of the conclusion. Author's should revise the conclusion section to provide conclusive statements on the research questions posed at the end of the introduction. The conclusion section should be revised.

Response to Comment 7: We have revised to conclusion to match the title and explain what we found in context of the community interventions:

Here. we have established criteria on social influences that lead to exercise persistence. Specifically, we found that the social interaction between moderately active individuals with the sedentary population should be engineered to draw the sedentary individuals into more active behaviors. A second criteria identified that strategies are needed to control dropouts from moderately active individuals. Finally, we found that discouraging socially acceptable sedentary behavior will reduce drop out of moderately active individuals. These findings can be used to design and implement community based exercise programs that lead to long-term exercise persistence in the community.

---

## [Decision Letter · Decision Letter 2]

25 Aug 2022

Social influences on physical activity for establishing criteria leading to exercise persistence

PONE-D-22-17305R2

Dear Dr. Thomas,

We’re pleased to inform you that your manuscript has been judged scientifically suitable for publication and will be formally accepted for publication once it meets all outstanding technical requirements.

Kind regards,

Zulkarnain Jaafar

Academic Editor

PLOS ONE

Additional Editor Comments (optional):

Reviewers' comments:

Reviewer's Responses to Questions

**Comments to the Author**

1. If the authors have adequately addressed your comments raised in a previous round of review and you feel that this manuscript is now acceptable for publication, you may indicate that here to bypass the “Comments to the Author” section, enter your conflict of interest statement in the “Confidential to Editor” section, and submit your "Accept" recommendation.

Reviewer #1: All comments have been addressed

2. Is the manuscript technically sound, and do the data support the conclusions?

Reviewer #1: Yes

3. Has the statistical analysis been performed appropriately and rigorously? 

Reviewer #1: Yes

4. Have the authors made all data underlying the findings in their manuscript fully available?

Reviewer #1: Yes

5. Is the manuscript presented in an intelligible fashion and written in standard English?

Reviewer #1: Yes

6. Review Comments to the Author

Reviewer #1: Based on the content of the latest revised manuscript, it is worth remarking that

a) the manuscript contains an interesting and novel aim,

b) the title is informative and relevant,

c) the introduction, literature review, methodology, results, discussion of results, conclusion and references are of high standard,

d) Author(s) have rigorously revised the manuscript. The present form of the whole report is also of high standard, and

e) the contribution of the report to the body of knowledge is significant.

Based on these aforementioned facts, it is worth concluding that the article is error free and suitable for publication. I hereby recommend "Acceptance". Congratulations to the authors for updating the body of knowledge with new scientific facts.

7. PLOS authors have the option to publish the peer review history of their article (what does this mean?). If published, this will include your full peer review and any attached files.

Reviewer #1: **Yes: **Dr. ANIMASAUN, Isaac Lare

---

## [Editor Report · Acceptance letter]

27 Sep 2022

PONE-D-22-17305R2 

Social influences on physical activity for establishing criteria leading to exercise persistence 

Dear Dr. Thomas:

I'm pleased to inform you that your manuscript has been deemed suitable for publication in PLOS ONE. Congratulations! Your manuscript is now with our production department. 

Kind regards, 

on behalf of

Dr. Zulkarnain Jaafar 

Academic Editor

PLOS ONE